# Peptide Toxins from Marine *Conus* Snails with Activity on Potassium Channels and/or Currents

**DOI:** 10.3390/toxins16120504

**Published:** 2024-11-22

**Authors:** Luis Martínez-Hernández, Estuardo López-Vera, Manuel B. Aguilar

**Affiliations:** 1Posgrado en Ciencias Biológicas, Instituto de Ciencias del Mar y Limnología, Universidad Nacional Autónoma de México, Ciudad de México 04510, Mexico; lamh14@gmail.com; 2Laboratorio de Toxinología Marina, Unidad Académica de Ecología y Biodiversidad Acuática, Instituto de Ciencias del Mar y Limnología, Universidad Nacional Autónoma de México, Ciudad de México 04510, Mexico; 3Laboratorio de Neurofarmacología Marina, Departamento de Neurobiología Celular y Molecular, Instituto de Neurobiología, Universidad Nacional Autónoma de México, Juriquilla 76230, Mexico

**Keywords:** conotoxins, voltage-gated potassium channels, conopeptides

## Abstract

Toxins from *Conus* snails are peptides characterized by a great structural and functional diversity. They have a high affinity for a wide range of membrane proteins such as ion channels, neurotransmitter transporters, and G protein-coupled receptors. Potassium ion channels are integral proteins of cell membranes that play vital roles in physiological processes in muscle and neuron cells, among others, and reports in the literature indicate that perturbation in their function (by mutations or ectopic expression) may result in the development and progression of different ailments in humans. This review aims to gather as much information as possible about *Conus* toxins (conotoxins) with an effect on potassium channels and/or currents, with a perspective of exploring the possibility of finding or developing a possible drug candidate from these toxins. The research indicates that, among the more than 900 species described for this genus, in only 14 species of the >100 studied to date have such toxins been found (classified according to the most specific evidence for each case), as follows: 17 toxins with activity on two groups of potassium channels (Kv and KCa), 4 toxins with activity on potassium currents, and 5 toxins that are thought to inhibit potassium channels by symptomatology and/or a high sequence similarity.

## 1. Introduction

Cone snails are marine gastropods grouped in the *Conus* genus; they are distributed mainly in tropical waters and possess a representative conical-shaped shell [1]. The study of *Conus* venom began in the mid-1960s [2]; however, it was not until 1978 that the biochemical characterization of the first toxins was described [3]. *Conus* toxins are peptides commonly classified into two structural groups, (i) those with two or more disulfide bridges and (ii) those with one or no disulfide bridges, named conotoxins and conopeptides, respectively [4,5]. Nevertheless, there are other molecules that are not included in this classification, as they have similarities with other proteins such as conkunitzins, which is similar to the bovine pancreatic trypsin inhibitor (Kunitz) or similar to α-dendrotoxin (*Dendroaspis angusticeps*), a potassium channel blocker [6].

At present, there are more than 900 living *Conus* species [7] from which toxins have been isolated, targeting a wide variety of membrane proteins such as voltage- and ligand-gated ion channels, neurotransmitter transporters, and G protein-coupled receptors [8]. As of 2022, only ten toxins from nine species were known to inhibit the activity of voltage-gated potassium channels (Kv) [9,10].

Potassium channels are the most complex class of voltage-gated ion channels, with 90 types, which are involved in cellular processes such as the regulation of resting membrane potential; the shape, duration, and frequency of action potentials; the modulation of neuronal excitability; and the release of neurotransmitters [11]. Based on the number and architecture of their alpha-subunits (transmembrane segments) that are arranged around a central pore and their activation mechanisms, they can be divided into (i) calcium-gated channels (KCa) with six or seven transmembrane segments and one pore, (ii) voltage-gated channels (Kv) with six transmembrane segments and one pore, (iii) channels with four transmembrane segments and double pores (K2P), and (iv) inward rectifier channels with two transmembrane segments and one pore (Kir) [12,13].

Peptide toxins that target Kv channels are denoted with the lowercase Greek letter Kappa (κ) [14] and have become a topic of interest due to the potential application they could have for the design of drugs that treat conditions in which Kv channels are involved [15,16].

Therefore, the purpose of this review is to provide researchers with information on conotoxins that affect potassium currents and/or channels, and also on several that are thought to act upon potassium currents, as well as the technique(s) used to demonstrate their pharmacological activities, thereby providing strong support for the advancement of new marine drug development efforts. Although an excellent review on conotoxins targeting calcium, sodium, and potassium channels was recently published [17], we include in ours not only toxins acting upon known subtypes of potassium channels and three examples that also inhibit sodium channels, but also conopeptides that affect potassium currents flowing through subtypes not defined with absolute certainty and those that elicit symptoms/behaviors considered to be associated with alterations in the function of diverse potassium channel subtypes. Finally, we also incorporate some peptides with a high sequence similarity to toxins with a proven effect on potassium channels and/or currents. We emphasize the species origin of the subtypes upon which conotoxins act, because we see that some of them affect a given subtype to different degrees in distinct species, which might have implications for the successful design of more potent/selective analogs.

## 2. Conotoxins

### 2.1. O-Conotoxins

The very first conotoxin discovered to have activity on potassium channels was κ-PVIIA, isolated from *Conus purpurascens*. This peptide of 27 amino acids in length belongs to the genetic superfamily O [18]. To use consistent nomenclature with respect to that used for other conotoxins discovered later, we followed the established nomenclature convention for toxins [14]; therefore, this toxin is formally designated as kO-PVIIA, where “O” indicates its membership in the O superfamily. Terlau and collaborators determined its activity on the *Shaker* potassium channel type by a two-electrode voltage clamp (TEVC) using heterologous expression in oocytes from *Xenopus laevis*. The concentration that produced 50% inhibition (IC_50_) was 60 nM, with no activity on rat Kv1.1 and Kv1.4 with a final concentration up to 1µM [18].

The lack of activity of κO-PVIIA on Kv1.1, which is the variant related to *Shaker* in mammals, caught the attention of researchers. Trying to explain this unexpected finding, in 1998, Shon and collaborators made chimeras transferring the S1–S4 part of the *Shaker* channel to Kv1.1 and vice versa (loop S5–S6), and 1 µM κO-PVIIA on *Shaker* did not block the currents of channels containing the loop of Kv1.1, while Kv1.1 with the *Shaker* loop became sensitive to κO-PVIIA. Based on these results, they concluded that the binding site of the toxin is in the pore-forming region [19]. Additional experiments showed that mutating the phenylalanine 425 to glycine and threonine 449 to tyrosine results in a channel that is insensitive to 1 µM κO-PVIIA, suggesting that this toxin is a pore-blocker [19].

In the same year, Dauplais and collaborators worked with pore-blocking peptide toxins for Kv isolated from sea anemones, BgK (*Bunodosoma granulifera*), ShK (*Stichodactyla helianthus*), AsKS (*Anemonia sulcata*), and charybdotoxin (*Leiurus quinquestriatus*) isolated from a scorpion. They proposed the existence of a functional dyad of amino acids responsible for their activity, which is composed of a positive amino acid (Lys) and a hydrophobic amino acid (Tyr or Phe) [20]. One year later, Savarin and collaborators carried out an analysis of the three-dimensional structure of κO-PVIIA and the possible interaction of each of the amino acids with Kv, suggesting that the mechanism is similar to that previously proposed by Dauplais and collaborators [21]. This hypothesis was corroborated in 2000 by Jacobsen and collaborators, carrying out the mutation of each of the amino acid residues of ĸO-PVIIA to alanine (except cysteine residues) and finding that the toxin affinity was due to three amino acids, Arg2, Lys7, and Phe9 [22].

Until 2000, data indicated that three amino acid residues were responsible for the activity of ĸO-PVIIA; however, the question of why κO-PVIIA blocks the *Shaker* channel and not the mammalian Kv1.1 remained unresolved. Two research teams took on the task of answering this question. Both teams resorted to computational modeling based on docking models and molecular dynamics (MD) simulations applied to protein–ligand complexes. Their results indicated that Lys7 of κO-PVIIA interacts with the Tyr445 of the *Shaker* loop and, at the same time, Phe9 acts closely with Phe425 [23,24]. Furthermore, the hydrogen bond cross-linking of Thr449 and Asp447 in *Shaker* allows the carbon side chain of Thr449 to form a hydrophobic interaction with the Phe9 of κO-PVIIA, favoring its binding [25]. When aligning the amino acid sequences of *Shaker* and rat Kv1.1, it was observed that, of the *Shaker* amino acids that allow interaction with κO-PVIIA (Phe425, Tyr445, Asp447, and Thr449), only two of them were conserved in Kv1.1 (*Shaker* Tyr445–Kv1.1 Tyr375 and *Shaker* Asp447–Kv1.1 Asp377) and two were not (*Shaker* Phe425–Kv1.1 His355 and *Shaker* Thr449–Kv1.1 Tyr379). Therefore, these authors proposed that the lack of affinity of κO-PVIIA for Kv1.1 is mainly due to differences at the structural level of the potassium channels in question.

The second conotoxin belonging to the O superfamily with activity on potassium channels was recently discovered from the venom of the snail *Conus spurius*. The κO-SrVIA conotoxin is 31 amino acids in length, and, using TEVC in *X*. *laevis* oocytes, it was reported to inhibit the human potassium channels Kv1.6, Kv10.1, and Kv11.1, with its IC_50_ being 3.6 µM, 1.88 µM, and 2.44 µM, respectively [26].

### 2.2. A-Conotoxins

Conotoxin kA-SIVA belongs to the A superfamily and is also known as “the spastic peptide”, because, when evaluated in fish, it causes spastic paralysis after an excitotoxic shock. It was identified by molecular biology techniques in the venom duct of *Conus striatus* and consists of 30 amino acids [27]. Using the TEVC technique, kA-SIVA caused a ≈50% decrease in the *Shaker* currents expressed in *X. laevis* oocytes when tested at a concentration of 2.5 µM [27]. However, subsequent experiments carried out by other authors demonstrated that this peptide, at concentrations higher than 30 nM, induced repetitive firing in the frog neuromuscular junction after a single stimulation; this effect was sensitive to TTX, which indicates that the toxin inhibits sodium channels (probably Nav1.6). In addition, no blockade of several Kv1 channels (*Shaker* B Δ6–46, *Xenopus* XKv1.1 and XKv1.3, and *Loligo* SqKv1.1) was observed at concentrations higher than those producing repetitive firing in the frog preparation. Also, κA-SIVA induced repetitive firing in frog nerves under tetraethylammonium at 12 mM, enough to block the [three] known types of Kv channels at the nodes of Ranvier of *Xenopus* [28]. These later findings strongly suggest that kA-SIVA’s primary mechanism of action is through sodium channel modulation rather than potassium channel blockade, and its initial classification as a κ-conotoxin may need to be reconsidered.

In 2004, Santos and collaborators reported a variant called kA-SIVB composed of 37 amino acid residues, 29 of which are the same as those present in kA-SIVA; therefore, it was proposed that they could share the same molecular target [29]. In the same research, when a toxin with 36 amino acid residues, MIVA, from the venom of *Conus magus*, was evaluated in fish, it elicited repetitive action potentials and caused the same spastic symptomatology as kA-SIVA [27]; thus, it was proposed that they could have the same molecular target [29]. These authors also reported two additional predicted sequences encoding 37-residue mature toxins, κA-SmIVA and κA-SmIVB (from *Conus stercusmuscarum*), that also have a considerable sequence similarity to κA-SIVA and κA-SIVB and, therefore, could be considered as potential κ-conotoxins [27,29]. However, since they have not been tested electrophysiologically, their effects on both potassium and sodium channels are an open question, considering the findings of Kelley and collaborators [28].

Conotoxins kA-PIVE and kA-PIVF are toxins composed of 24 amino acid residues and were isolated by milking the venom of *Conus purpurascens*. Bioassays performed in goldfish with these two conotoxins have indicated an excitotoxic effect similar to that reported with the kA-SIVA conotoxin [30]. However, only the synthetic variant of kA-PIVE has been shown to have activity on potassium currents in the skeletal muscle of frogs at a concentration of 10 µM [30].

### 2.3. M-Conotoxins

Conotoxin κM-RIIIK is one of the most studied toxins that acts on potassium channels, after κO-PVIIA. This toxin belongs to the genetic superfamily M; it was isolated from the venom of *Conus radiatus* and has 24 amino acids in its primary structure [31]. TEVC studies in *X. laevis* oocytes determined that it blocks *Shaker* channels with an irreversible effect and an IC_50_ of 1.21 µM [31]. Furthermore, it has been suggested that Lys427 of the *Shaker* channel is responsible for the binding of the toxin, since when this amino acid is substituted by Asp or Asn, the IC_50_ increases by more than 100 times. The same study uncovered that κM-RIIIK has no effect on other potassium channels such as Kv1.1, Kv1.3, Kv1.4, Kv2.2, Kv3.4, Kv4.2, herg, and r-eag, nor on the rat sodium channel Nav1.4 at a concentration of 10 µM [31].

A year later, it was documented that κM-RIIIK irreversibly blocks the *TSha1* channel of the rainbow trout (*Oncorhynchus mykiss*) with an IC_50_ of 73 nM, and the amino acids responsible for this affinity are Leu1, Arg10, Lys18, and Arg19, indicating that κM-RIIIK inhibition is due to a pharmacophore formed by a ring of positively charged amino acids that differs from that of κO-PVIIA [32]. Subsequent studies showed that the half-inhibitory concentrations in human and rat Kv1.2 were 352 nM, and 335 nM, respectively, and there was no inhibitory activity on the rat subtypes Kv1.1, Kv1.5, and Kv1.6 or human subtypes Kv1.2-Kv1.6, KCNQ2/KCNQ3, and BK (KCa) at a final concentration of 5 to 10 µM [33]. This is a clear example of a toxin that has considerably different affinities for homologous subtypes from two different species.

κM-RIIIJ conotoxin was isolated from the venom of the same species as κM-RIIIK (*C. radiatus*), and this toxin is 25 amino acids in length. TEVC revealed that it blocks the rat subtype Kv1.1 and human subtypes Kv1.2, Kv1.3, Kv1.5, and Kv1.6 (IC_50_ 4 µM, 33 nM, 10 µM, 70 µM, and 8 µM, respectively) [34]. Chen and collaborators also showed that Lys9 residue is responsible for its affinity for Kv1.2 and that, even at concentrations higher than 70 µM, no significant inhibitory activity was observed on the human subtypes Kv1.4, KCNQ2/KCNQ3, and KCa [34]. In 2019, Cordeiro and colleagues expressed heterodimeric or asymmetric subtypes of human channels in HEK293 cells, with combinations of Kv1.2 and two other subtypes. Using a whole-cell patch clamp, their results indicated that the affinity of κM-RIIIJ increased 100-fold for the channels formed by combinations of three subunits of Kv1.2 and one subunit of Kv1.1 or Kv1.6 (3:1), with respect to the homomeric channels [35].

The Vt3.1 toxin was identified from the venom of *Conus vitulinus*, and it belongs to superfamily M. However, its classification as a conotoxin is still pending; structurally, it has two chains of 13 amino acids each (dimer) [36]. Its activity on potassium channels was determined by a patch clamp (outside–out) in *X. laevis* oocytes expressing the rat α-mslo1 subunit (KCa) alone or together with human beta subunits (β1, β2, β3, and β4). The results showed that Vt3.1 irreversibly blocked mslo1 with β4 with a IC_50_ of 8.5 µM, with no inhibition higher than 30% at a 10 µM final concentration on mslo1 + β1, mslo + β2, and mslo + β3 [37]. Furthermore, the existence of a Vt3.2 variant that differs from Vt3.1 in cysteine disulfide connectivity and has no activity on mslo+ β4 at a concentration of 10 µM has been reported [37].

Conotoxin μM-SIIIA is a 20-aa peptide isolated from *Conus striatus* that potently blocks the neuronal rNav1.2, rNav1.4, and mNav1.6 subtypes, and, to a lesser degree, hNav1.7; it does not block the cardiac hNav1.5 channels. However, its synthetic form, at 10 µM, reduces the current of rKv1.1 by 27.3% and that of mKv1.6 by 54.0%, with very small effects on the rKv1.2, mKv1.3, rKv1.4, and hKv1.5 subtypes and rKv2.1 [38]. In the same report, it was demonstrated that conotoxin μ-PIIIA (22 aa, from *C. purpurascens*), a sodium channel blocker that clearly inhibits the rNav1.4 muscle sodium channel and the neuronal subtypes rNav1.2 and hNav1.7, also reduces the currents, at 10 µM, of rKv1.1 (73.0%) and mKv1.6 (80.3%), but not those of rKv1.2, mKv1.3, rKv1.4, and hKv1.5 or rKv2.1. Studies of the effects of this toxin on diverse chimeras of hKv1.5 and mKv1.6 indicated that the blocking of Kv channels involves an interaction with their pore regions; the preference for particular subtypes is determined partially by the sequence adjacent to the selectivity filter, but mainly by the turret domain [38].

In 2021, a toxin belonging to the superfamily M from *Conus varius* was isolated and named Vr3a. This peptide is 21 aa in length, 6 of which are prolines. Its activity was evaluated on neurons from the rat dorsal root ganglion (DRG), showing an increase of around 20% in the potassium current at a 10 µM final concentration [39].

### 2.4. J-Conotoxins

The κJ superfamily includes the PIXIVA (pl14a) conotoxin, which comprises 25 amino acid residues and was isolated from the venom of *Conus planorbis.* TEVC in *X*. *laevis* oocytes that expressed potassium channels showed that PlXIVA has an IC_50_ of 1.59 µM for the Kv1.6 channel, while at concentrations of 1 µM, a low percentage of Kv1.1 blockage is reported and no activity in Kv1.2-Kv1.5 was detected; likewise, no inhibition of the currents of the Kv2.1, Kv3.4, and Nav1.2 channels was reported at a concentration of 2 µM. Surprisingly, when evaluating this toxin in nicotinic acetylcholine receptors (nAChRs), the result was inhibition in the rat α3β4 and mouse α1β1εδ subtypes with an IC_50_ of 8.7 µM and 0.54 µM, respectively [40].

### 2.5. I_2_-Conotoxins

In 2003, two conotoxins of the I_2_ superfamily with activity on potassium channels were isolated, ViTx (33 amino acid residues) and BtX (31 amino acid residues). The first of them was isolated from the venom of *Conus virgo*, and TEVC in *X*. *laevis* oocytes showed that it causes an irreversible inhibition of the rKv1.1 and hKv1.3 channels with dissociation constants (Kd) of 1.59 µM and of 2.09 µM, respectively. Furthermore, there was no effect on the rKv1.2, rKir 2.1, and herg channels [41].

The second toxin, BtX, was isolated from the venom of *Conus betulinus.* Using a patch clamp (whole-cell) in rat adrenal chromaffin cells (RACCs), a twofold increase in potassium currents was observed at 1 µM. In the same investigation, its activity on high-conductance rat KCa channels was determined, reporting a mean effective concentration (EC_50_) of 0.7 nM, inducing an increase in the amplitude of the currents, in addition to favoring the probability and opening time of these channels [42].

A third conotoxin of the I_2_ superfamily was isolated from the venom of *C. spurius*, SrXIA. This toxin consists of 32 amino acid residues and has a sequence similarity of 51% with BtX and 31% with ViTx. Using TEVC in *X*. *laevis* oocytes, with a concentration of 640 nM, a slow inhibitory activity was determined (14–23 min), being 66% in rat Kv1.2, 58% in human Kv1.6, and without activity in rat Kv1.3 [43,44].

## 3. Conopeptides

### 3.1. Contryphan-Vn

Contryphan-Vn was isolated from the venom of *Conus ventricosus* and it is considered to be one of the smallest components that has been reported in *Conus* venom. Structurally, it is composed of nine amino acid residues and has a single disulfide bridge [45,46]. Its activity on potassium currents was determined by a patch clamp (whole-cell) in unpaired dorsal median neurons (DUM) of the American cockroach *Periplaneta americana* and in rat chromaffin cells. The evaluation of this toxin at a concentration of 20 µM produces an increase in the frequency of action potential discharge in DUMs and an increase in the potassium currents for chromaffin cells [47]. This last effect was similar to that reported with BtX.

### 3.2. Mo1659

Conopeptide Mo1659, named for its monoisotopic molecular mass (1659.1 daltons), was isolated from the venom of *Conus monile.* This peptide does not have disulfide bridges, and structurally, it consists of 13 amino acid residues and is characterized by having 6 aromatic amino acid residues (Phe1, Phe9, Tyr7, Tyr13, Trp6, and Trp11) [48]. Electrophysiological recording in the DRG of rats using a patch clamp (whole-cell) indicated that, with a concentration of 200 nM, its application results in a reduction of more than 50% in potassium currents.

A year later, Kumar and collaborators reported that the biosynthetic variant, which is characterized by lacking C-terminal amidation, when evaluated at a concentration of 200 nM in DRG neurons by patch clamp (whole cell), they saw an inhibition ≈ 30% in potassium currents. Something important to highlight is that the inhibitory activity can increase up to 50% if potassium currents are generated followed by a pre-pulse to inactivate the potassium channels present in these cells [49], the same as reported with natural or wild-type Mo1659 [48]. The enhanced inhibition following an inactivating pre-pulse suggests a state-dependent binding of the toxin, with a potentially higher affinity for the inactivated state of the channels.

### 3.3. CPY Peptides

Conopeptides CPY-Fe1 and CPY-Pl1, isolated from the snails *Conus ferrugineus* and *Conus planorbis*, respectively, are 30 amino acid residues long and are characterized by a high content of tyrosine residues. Their potassium channel activity was determined by TEVC in *X*. *laevis* oocytes that expressed mammalian potassium channels. For CPY-Fe1, an IC_50_ of 8.8 µM was reported in Kv1.6, without presenting activity in the Kv1.2, Kv1.3, Kv1.4, and Kv1.5 channels at a concentration of ≥ 30 µM. With respect to CPY-Pl1, an IC_50_ of 170 nM for Kv1.6 and an IC_50_ of 2 µM for Kv1.2 were determined, respectively. Similarly to CPY-Fe1, when the activity of CPY-Pl1 in the Kv1.3, Kv1.4, and Kv1.5 channels was eval-uated, no significant activity was observed at a concentration of ≥ 5 µM [50].

### 3.4. Conorfarmide-Sr3

Conorfamide CNF-Sr3 was isolated from the venom of *Conus spurius*, and it receives this name because it is closely related to the RF-amide neuropeptides of mollusks [51,52]. Structurally, it consists of 15 amino acid residues, and, by a patch clamp in sf9 cells transfected with *Shaker,* a Kd of 2.7 µM was reported. Evaluation at concentrations between 5 and 10 µM did not produce significant inhibitory activity in *Shab* (sf9 cells), *Shaw*, *Shal,* and *eag* (HEK-293 cells) [53]. Three years later, experiments carried out by TEVC in *X*. *laevis* oocytes showed a significant inhibitory activity in human Kv1.6 and Kv1.3 channels (IC_50_ 2.7 µM and 24 µM, respectively) and did not present significant inhibitory activity in the human Kv1.4 and Kv1.5 channels at a concentration of up to 10 µM [54].

## 4. Conkunitzin

Conkunitzin-S1 (Conk-S1) was identified in the venom of *Conus striatus*, and it is considered as one of the largest components in *Conus* venom, since it is composed of 60 amino acid residues in its primary structure [6]. Conk-S1 has not been related to any of the conotoxin superfamilies and, in fact, is more related to proteins containing the Kunitz domains, such as α-dendrotoxin, with which it shares 42% identity [55]. However, unlike α-dendrotoxin, which has three disulfide bridges, Conk-S1 only has two [6].

Its activity on *Shaker*-type (∆6–46) potassium channels was determined by TEVC in *X*. *laevis* oocytes, presenting an IC_50_ of 502 nM. Similarly, a variant of Conk-S1, called Conk-S1^cc^, which is characterized by having an additional disulfide bridge, as present in α-dendrotoxin, was evaluated in *Shaker*-∆6-46, and, surprisingly, the affinity of this toxin increased, exhibiting an IC_50_ of 385 nM. In 2012, Finol-Urdaneta and collaborators determined by patch clamp (whole-cell) that, at a concentration of 1 µM, Conk-S1 reversibly blocked more than 50% of the currents of the expressed the human Kv1.7 channel in tsA-201 cells, and that in murine Kv1.7, it had an IC_50_ of 439 nM, without having an effect on other human (Kv1.1, Kv1.3–1.6, Kv2.2, and Kv4.2) and rat (Kv2.1, Kv3.1, Kv3.2, Kv3.4, reag1, and reag2) potassium channels at concentrations of ≥30µM [56].

Finally, using molecular biology techniques, conkunitzin-C3 (Conk-C3) was isolated from the venom duct of *Conus consors* [57]. This toxin, in addition to having the same length of amino acid residues as Conk-S1, also shares a similarity of 86% of its sequence and the same molecular target (*Shaker*). The evaluation of Conk-C3 at 500 nM by TEVC in *X*. *laevis* oocytes indicated an inhibitory activity close to 50%. However, molecular dynamics simulations proposed that its mechanism of interaction with the channel is different from that of Conk-S1 [57].

To summarize, all the natural conotoxins with activity on the potassium channels and/or currents described above are contained in Table 1 and Figure 1.

## 5. Discussion

The study of potassium channels has become a topic addressed by multiple research groups, given the history of their participation in a wide variety of conditions in humans, which range from heart problems, epilepsy, and childhood development disorders to cancer [16,59,60,61]. To understand the participation of potassium channels in these conditions, blockers such as 4-aminopyridine (4-AP) and tetraethylammonium (TEA) have been used; however, the scope provided by these molecules is limited, because their activity is restricted to blocking the pore of channels [62]. Having said that, the search for peptide toxins that can modulate the activity of these channels has been a topic of great relevance [63,64], since their activity is not limited to blocking the pore, and they are capable of interacting with the voltage sensor domain and, thus, modulating the gating of the Kv. Such is the case of the peptide toxin hanatoxin (HaTx), isolated from the tarantula *Grammostola spatulate* [65]. Since the biochemical characterization of the first *Conus* toxin [3], the study of these toxins has increased significantly; most studies have had the objective of using electrophysiological techniques in order to determine the molecular target for which they have affinity. In this way, by finding a conotoxin that presents specificity for a molecular target, it is possible to attribute a potential application for drug design. The best-known case is the ω-conotoxin MVIIA, a peptide of 25 amino acid residues isolated from the venom of *Conus magus*, which has a relevant affinity for mammalian Cav2.2 calcium channels. Currently, its synthetic variant, known as Ziconotide or Prialt (SNX-111, US Patent 5364842), is used for the treatment of chronic pain in patients with cancer and AIDS [66,67]. Regarding *Conus* toxins that target potassium channels, multiple reviews have focused on gathering information on their biochemical and electrophysiological characteristics (for more details see references [8,68,69]). The most recent information available indicates that around 10 toxins with activity in voltage-activated potassium channels have been characterized [9,10,17].

Among the conotoxins that inhibit Kv, the most outstanding are κO-PVIIA, κM-RIIIK, and κM-RIIIJ, since they are the most studied, and it has been reported that their activity can vary depending on the conformational state of the Kv and changes in affinity related to whether the toxin is evaluated in homomeric or heteromeric channels [35,36,70]. The characteristics of the toxins mentioned above have allowed them to be candidates in in vivo tests. For example, in ischemia and reperfusion models, κM-RIIIJ has the greatest cardioprotective effect with respect to the other two conotoxins [34,71,72].

The κM-RIIIJ conotoxin has attracted the most attention in recent years, given its property of binding to heteromeric Kv channels, and that, along with calcium imaging records, has allowed us to consider it as an important molecule to identify neuronal subclasses (L1 and L2) [for more details see reference [73], a characteristic that has been highlighted to be of great importance when studying diseases such as multiple sclerosis [62].

Taking into consideration that the Kv1.2, Kv1.3, and Kv1.6 subtypes are the main targets of most conotoxins and conopeptides (Table 2), it is reasonable to assume that they could be involved during the capture or defense behaviors in organisms in their natural habitats and, even more importantly, that a possible usefulness could be assumed in the study of some conditions in which these subtypes are related, such as the case of Parkinson’s disease, where it has been reported that they are involved in the progress of said disease [74]. Another important case to highlight is the activity of conotoxins on KCa (slo1 subunit), since mutations in these channels have been associated with epilepsy and with the “fragile X chromosome”, which modulates presynaptic activity by interaction with the β4 subunit [37].

With respect to the conkunitzin group, of the two natural members known to date with activity in potassium channels, Conk-S1 is a case of special attention, given its activity in mKv1.7, which has been proposed to be related to the increase in insulin secretion by pancreatic cells when the toxin is evaluated in rats [9,56]. A salient fact related to this group of *Conus* toxins is that, until 2019, they were thought to act in a similar way to pore-blocking toxins (“plug”). However, research carried out in the last four years has allowed for hypotheses that conkunitzins can interact in two additional ways, causing pore collapse, resulting in molecular effects similar to the slow inactivation of Kv channels [75] or as a lid mechanism (“molecular-lid”), as proposed for Conk-C3 [57]. The latter is of great relevance due to the importance of identifying molecules with mechanisms that could modify the activity of potassium channels and, thus, take advantage of this property for the design of therapeutics based on peptide toxins, as has been highlighted [63].

Finally, a conotoxin recently reported by our work team, κO-SrVIA, is the first toxin identified in any *Conus* venom that has the potassium channels Kv1.6, Kv10.1, and Kv11.1 as affected molecular targets [27]. These subtypes have been involved in the development of cardiac channelopathies and, even more importantly, in tumor development [16,59,60,76,77]. Taking the latter into account, it is important to highlight the potential that κO-SrVIA can present, since, when compared with other peptide toxins with which it shares Kv10.1 as a molecular target, κO-SrVIA brings together several of its more relevant structural and functional characteristics [27].

## 6. Conclusions

This review was focused on gathering the greatest amount of information on natural *Conus* snail toxins with relevant affinity for potassium channels and/or currents (Table 1). As of August 2024, there is a record of 17 *Conus* toxins that present activity on potassium channels (PVIIA, SrVIA, SIVA, RIIIK, RIIIJ, Vt3.1, PlXIVA, ViTx, BtX, SrXIA, CPY-Fe1, CPY-Pl1, CNF-Sr3, Conk-S1, Conk-C3, PIIIA, and SIIIA), four with activity on potassium currents (PIVE, Vr3a, Contryphan-Vn, and Mo1659), two that present similar effects as kA-SIVA when evaluated in goldfish (MIVA, repetitive action potentials and spastic symptomatology, and PIVF, excitotoxic effect), and three (SIVB, SmIVA, and SmIVB) that, due to a high degree of sequence similarity to SIVA, are thought to have activity in potassium channels.

## Figures and Tables

**Figure 1 toxins-16-00504-f001:**
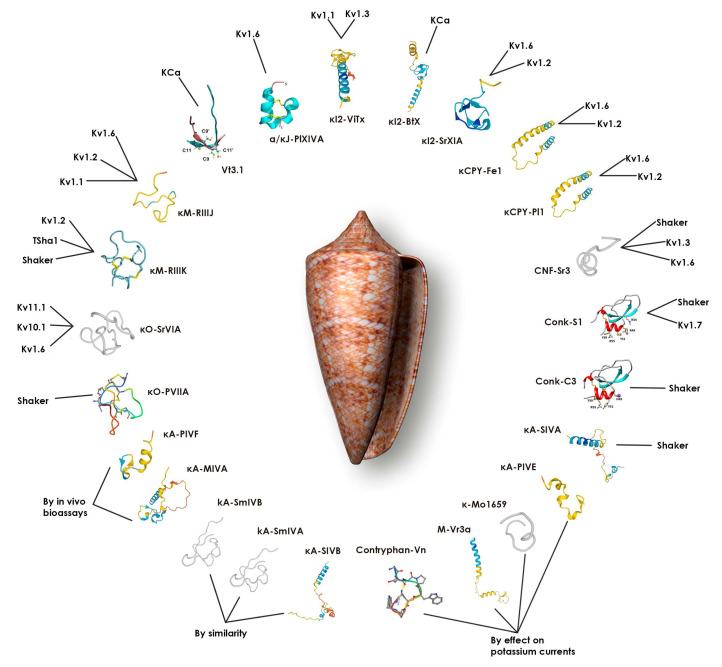
Comprehensive overview of potassium channel-targeting conotoxins and their structural/functional relationships. The figure uses a radial organization around a central *Conus* shell, presenting information in three concentric layers: Inner layer, names of the toxins. Middle layer, structural information: conotoxin structures are shown as ribbon diagrams; experimentally determined structures (Protein Data Bank) are displayed in the color scheme that appears in the thumbnail images of each toxin, while computational predictions (PEP-FOLD3) appear in gray. Outer layer, evidence classification: Target subtype (connecting lines indicate validated interactions with specific potassium channel targets; multiple connections highlight toxins with broad-spectrum activity), “By in vivo bioassays” (Toxins validated through functional studies), “By similarity” (candidate toxins identified through sequence homology analysis), or “By effect on potassium currents” (toxins with demonstrated electrophysiological effects).

**Table 1 toxins-16-00504-t001:** Peptide toxins isolated from marine *Conus* species with activity against potassium channels ^a^ currents ^b^ and thought to affect them by symptomatology ^c^ when injected or by sequence similarity ^d^. Sequences were retrieved from ConoServer [58].

Species	Toxin	Sequence
*C. purpurascens*(P)	κO-PVIIA ^a^	CRIONQKCFQHLDDCCSRKCNRFNKCV-NH2
κO-PIVE ^b,c^	DCCGVKLEMCHPCLCDNSCKNYGK-NH2
κO-PIVF ^c,d^	DCCGVKLEMCHPCLCDNSCKKSGK-NH2
*C. spurius*(V)	κI_2_-SrXIA ^a^	CRTEGMSCγγNQQCCWRSCCRGECEAPCRFGP-NH2
CNF-Sr3 ^a^	ATSGPMGWLPVFYRF-NH2
κO-SrVIA ^a^	GCGVDGQFCGLPGLGLVCCRGACFLVCIYIP
*C. radiatus*(P)	κM-RIIIK ^a^	LOSCCSLNLRLCOVOACKRNOCCT-NH2
κM-RIIIJ ^a^	LOOCCTOOKKHCOAOACKYKOCCKS
*C. vitulinus*(V)	Vt3.1 ^a^	GPYRRYGNCYCPI-NH2 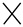 GPYRRYGNCYCPI-NH2
*C. planorbis*(V)	α/κJ-PlXIVA ^a^	FPRPRICNLACRAGIGHKYPFCHCR-NH2
κ-CPY-Pl1 ^a^	ARFLHPFQYYTLYRYLTRFLHRYPIYYIRY
*C. virgo*(V)	κI_2_-ViTx ^a^	SRCFPPGIYCTPYLPCCWGICCGTCRNVCHLRI
*C. betulinus*(V)	κI_2_-BtX ^a,b^	CRAγGTYCγNDSQCCLNγCCWGGCGHOCRHP-NH2
*C. ferrugineus*(V)	κCPY-Fe1 ^a^	GTYLYPFSYYRLWRYFTRFLHKQPYYYVHI
*C. striatus*(P)	κA-SIVA ^a,c^	ZKSLVP(gSr)VITTCCGYDOGTMCOOCRCTNSC-NH2
κA-SIVB ^d^	ZKELVP(gSr)VITTCCGYDOGTMCOOCRCTNSCOTKOKKO-NH2
Conk-S1 ^a^	KDRPSLCDLPADSGSGTKAEKRIYYNSARKQCLRFDYTGQGGNENNFRRTYDCQRTCLYT
*C. consors*(P)	Conk-C3 ^a^	DRPSYCNLPADSGSGTKSEQRIYYNSARKQCLTFTYNGKGGNENNFIHTYDCRRTCQYPA
*C. monile*(V)	Mo1659 ^b^	FHGGSWYRFPWGY-NH2
*C. varius*(V)	M-Vr3 ^b^	QGCCPPGVCQMAACNPPPCCP
*C. ventricosus*(V)	Contryphan-Vn ^b^	GDCPWKPWC-NH2
*C. magus*(P)	κA-MIVA ^c^	AOγLVV(gTr)A(gTr)TNCCGYNOMTICOOCMCTYSCOOKRKO-NH2
*C. stercusmuscarum*(P)	κA-SmIVA ^d^	ZTWLVP(gSr)(gTr)ITTCCGYDOGTMCOTCMCDNTCKOKOKKS-NH2
*C. stercusmuscarum*(P)	κA-SmIVB ^d^	ZPWLVP(gSr)(gTr)ITTCCGYDOGSMCOOCMCDNNTCKOKOKKS-NH2

The one-letter code is employed for the standard amino acids; NH-2, C-terminal amidation; O, 4-hydroxyproline; γ, gamma carboxyglutamate; Z, pyroglutamate; W, 6-bromotryptophan; gTr, threonine-glucosylation; gSr, serine-glucosylation; 
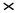
, denotes disulfide connectivity. Letter between parentheses below species name indicates type of feeding P; piscivorous, V; vermivorous.

**Table 2 toxins-16-00504-t002:** Peptide toxins isolated from marine *Conus* species targeting voltage-gated potassium (Kv) channel subtypes.

Toxin	Target Potassium Channel(s) (*IC_50_, °Kd, ^EC_50_)	References
κO-PVIIA	**Shaker* (60 nM)	[18]
κO-SrVIA	*hKv1.6 (3.6 µM); *hKv10.1 (1.88 µM); *hKv11.1 (2.44 µM)	[26]
κM-RIIIK	**Shaker* (1.21 µM); **TSha1* (73 nM); *hKv1.2 (352 nM); *rKv1.2 (335 nM)	[32,33,34]
κM-RIIIJ	*hKv1.2 (33 nM); *hKv1.3 (10µM); *hKv1.5(70µM); *hKv1.6 (8 µM); *rKv1.1 (4 µM)	[35]
ĸA-SIVA	**Shaker* (≈50% current decrease at 2.5 μM)	[27]
Vt3.1	*rat α-mslo1 + hβ4 (8.5 µM)	[38]
α/κJ-PlXIVA	*Kv1.6 (1.59 µM)	[40]
κI_2_-ViTx	°rKv1.1 (1.59 µM); °hKv1.3 (2.09 µM)	[41]
κI_2_-BtX	^KCa (0.7 nM)	[42]
κI_2_-SrXIA	*rKv1.2 (66% current decrease at 640 nM); *hKv1.6 (58% current decrease at 640 nM)	[44]
κ-CPY-Pl1	*Kv1.6 (170 nM); *Kv1.2 (2 µM)	[50]
Κ-CPY-Fe1	*Kv1.6 (8.8 µM)	[50]
CNF-Sr3	°*Shaker* (2.7 µM); *hKv1.6 (2.7 µM); *hKv1.3 (24 µM)	[53,54]
Conk-S1	**Shaker*-∆6–46 (502 nM); *mKv1.7(439 nM)	[6,56]
Conk-C3	**Shaker* (50% current decrease at 500 nM)	[57]

## Data Availability

No new data were created or analyzed in this study. Data sharing is not applicable to this article.

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
