# Peer review of "Peptide Toxins from Marine Conus Snails with Activity on Potassium Channels and/or Currents"

_toxins, 2024, doi:10.3390/toxins16120504_

Round 1
Reviewer 1 Report
Comments and Suggestions for Authors
In this manuscript, conotoxins that affect potassium channels or potassium currents are summarized. It is benefit for us to understand the structure, activity and molecular mechanisms of potassium channel targeted conotoxins. The expression is clear and professional. It is suitable for publication in this journal.
Nevertheless, minor modifications are still needed.
1) In Table 1, Two same sequences are given for Vt3.1 from the species of C.vitulinus, please modify.
2) In P9, line 344, “……of the first Conus toxin3,…..”, please given the meaning of the superscript number 3.
3) In P11, line 410, “two that present similar effects when …..”, please give the detailed effects about “similar effects”.
4) The format of the Reference is inconsistent, for example: P12, Line431, Reference 5, “Akondi K. B.; et al.”, in order to consistent with other references style, please give the name of other authors to replace “et al.”
5) It will be better to use three-line table style in Table1 and Table2.
Author Response
Comment: In this manuscript, conotoxins that affect potassium channels or potassium currents are summarized. It is benefit for us to understand the structure, activity and molecular mechanisms of potassium channel targeted conotoxins. The expression is clear and professional. It is suitable for publication in this journal.
Nevertheless, minor modifications are still needed.
- In Table 1, Two same sequences are given for Vt3.1 from the species of C. vitulinus, please modify.
Answer: The sequences correspond to a dimeric toxin. The connectivity (displayed as a big “X”) between cysteine residues changed at the moment to upload the document. We have done the correction.
2) In P9, line 344, “……of the first Conus toxin3,…..”, please given the meaning of the superscript number 3.
Answer: Thank you very much for your observation; it must be a reference instead of a superscript number. The observation was corrected.
3) In P11, line 410, “two that present similar effects when …..”, please give the detailed effects about “similar effects”.
Answer: Thank you for the remark; we have given the detailed effects.
4) The format of the Reference is inconsistent, for example: P12, Line431, Reference 5, “Akondi K. B.; et al.”, in order to consistent with other references style, please give the name of other authors to replace “et al.”
Answer: Thank you very much again; we have corrected the reference.
5) It will be better to use three-line table style in Table1 and Table2.
Answer: Thank you for your suggestion; we decided to keep the format specified in the template of the journal.
Reviewer 2 Report
Comments and Suggestions for Authors
This review provides a valuable and focused compilation of current knowledge about peptide toxins from marine Conus snails that target potassium channels. The authors set out to help readers quickly understand how κ-conotoxins work, elucidate the techniques used for target identification, summarize their binding affinities, and explore the relationship between prey preference and toxin expression. These objectives are largely met through systematic presentation of the data. The comprehensive coverage of different toxin superfamilies, coupled with detailed molecular characterizations, are notable and valuable. However, several points require attention before publication to enhance clarity and precision:
- Nomenclature explanation: The statement about kO-PVIIA nomenclature should acknowledge the existing convention rather than implying a new approach. I suggest rephrasing to: "Following the established nomenclature convention for conotoxins (King et al., 2008 (14)), this toxin is formally designated as kO-PVIIA, where 'O' indicates its membership in the O superfamily."
- Conclusive analysis: Some sections presenting contradictory findings would benefit from clear conclusions. For example, the kA-SIVA section could conclude with: "These later findings strongly suggest that kA-SIVA's primary mechanism of action is through sodium channel modulation rather than potassium channel blockade, and its initial classification as a κ-conotoxin may need to be reconsidered."
- Review-appropriate language: Several sentences need rephrasing to better reflect review paper conventions. For instance, "In the κJ-Superfamily we found..." should be "The κJ-Superfamily includes..."
- Mechanistic insights: Some experimental findings would benefit from explicit interpretation. For example, for the Mo1659 prepulse effect description I suggest to add "The enhanced inhibition following an inactivating prepulse suggests state-dependent binding of the toxin, with potentially higher affinity for the inactivated state of the channels."
- Figure 1 needs revision. The fonts are two small in the printed version, and the legend is too simplistic for the amount of information the authors wish to convey. I suggest something like :
Figure 1: Comprehensive overview of potassium channel-targeting conotoxins and their structural/functional relationships. The figure uses a radial organization around a central Conus shell, presenting information in three concentric layers:
A) Structural Information: Conotoxin structures are shown as ribbon diagrams. Experimentally determined structures (Protein Data Bank) are displayed in color, while computational predictions (PEP-FOLD3) appear in gray. [Note: The color scheme for structural elements should be explicitly defined]
B) Evidence Classification:
- "By in vivo bioassays": Toxins validated through functional studies
- "By similarity": Candidate toxins identified through sequence homology analysis
- "By effect on potassium currents": Toxins with demonstrated electrophysiological effects
C) Target Specificity: Connecting lines indicate validated interactions with specific potassium channel targets (Kv1.1-1.7, KCa, Shaker, etc.). Multiple connections highlight toxins with broad-spectrum activity."
In summary, this review successfully achieves its primary goal of providing a comprehensive yet focused overview of Conus peptide toxins targeting potassium channels. I recommend publication following attention to the points above.
Author Response
Comment: This review provides a valuable and focused compilation of current knowledge about peptide toxins from marine Conus snails that target potassium channels. The authors set out to help readers quickly understand how κ-conotoxins work, elucidate the techniques used for target identification, summarize their binding affinities, and explore the relationship between prey preference and toxin expression. These objectives are largely met through systematic presentation of the data. The comprehensive coverage of different toxin superfamilies, coupled with detailed molecular characterizations, are notable and valuable. However, several points require attention before publication to enhance clarity and precision:
- Nomenclature explanation: The statement about kO-PVIIA nomenclature should acknowledge the existing convention rather than implying a new approach. I suggest rephrasing to: "Following the established nomenclature convention for conotoxins (King et al., 2008 (14)), this toxin is formally designated as kO-PVIIA, where 'O' indicates its membership in the O superfamily."
Answer: Thank you very much; we have rephrased the text according to your observation.
- Conclusive analysis: Some sections presenting contradictory findings would benefit from clear conclusions. For example, the kA-SIVA section could conclude with: "These later findings strongly suggest that kA-SIVA's primary mechanism of action is through sodium channel modulation rather than potassium channel blockade, and its initial classification as a κ-conotoxin may need to be reconsidered."
Answer: Thank you very much again; we have included the text according to your suggestion.
- Review-appropriate language: Several sentences need rephrasing to better reflect review paper conventions. For instance, "In the κJ-Superfamily we found..." should be "The κJ-Superfamily includes..."
Answer: Thank you so much; We have reviewed sentences throughout the manuscript and made changes based on your observation.
- Mechanistic insights: Some experimental findings would benefit from explicit interpretation. For example, for the Mo1659 prepulse effect description I suggest to add "The enhanced inhibition following an inactivating prepulse suggests state-dependent binding of the toxin, with potentially higher affinity for the inactivated state of the channels."
Answer: Thank you very much again; we have included the text according to your suggestion.
- Figure 1 needs revision. The fonts are two small in the printed version, and the legend is too simplistic for the amount of information the authors wish to convey. I suggest something like :
Figure 1: Comprehensive overview of potassium channel-targeting conotoxins and their structural/functional relationships. The figure uses a radial organization around a central Conus shell, presenting information in three concentric layers:
- A) Structural Information: Conotoxin structures are shown as ribbon diagrams. Experimentally determined structures (Protein Data Bank) are displayed in color, while computational predictions (PEP-FOLD3) appear in gray. [Note: The color scheme for structural elements should be explicitly defined]
- B) Evidence Classification:
- "By in vivo bioassays": Toxins validated through functional studies
- "By similarity": Candidate toxins identified through sequence homology analysis
- "By effect on potassium currents": Toxins with demonstrated electrophysiological effects
- C) Target Specificity: Connecting lines indicate validated interactions with specific potassium channel targets (Kv1.1-1.7, KCa, Shaker, etc.). Multiple connections highlight toxins with broad-spectrum activity."
Answer: Thank you; tentatively, we kept the font size to avoid crowing of the texts in the figure. We have included the text according to your suggestion, with slight changes.
In summary, this review successfully achieves its primary goal of providing a comprehensive yet focused overview of Conus peptide toxins targeting potassium channels. I recommend publication following attention to the points above.
Answer: Thank you very much for your comment.
Reviewer 3 Report
Comments and Suggestions for Authors
I feel the paper needs to be rewritten for a wider audience.
Comments on the Quality of English LanguageIn its current form I regret that I would reject the current manuscript. The manuscript is hard to read in its current format. Expression and grammar are poor throughout.
Author Response
Comments and Suggestions for Authors
I feel the paper needs to be rewritten for a wider audience.
Answer: We would appreciate if you could give us further explanation about “wider audience”, because, it is a scientific paper.
Comments on the Quality of English Language
In its current form I regret that I would reject the current manuscript. The manuscript is hard to read in its current format. Expression and grammar are poor throughout.
Answer: We have done some changes in the manuscript as the other reviewers also recommended improving the English.
Round 2
Reviewer 3 Report
Comments and Suggestions for Authors Happy with the current version.Comments on the Quality of English Language Happy with the current version.